# Position: AI Evaluation Should Work With Humans

Jan Kulveit [1]   Gavin Leech [2]   Tomáš Gavenčiak [1]   Raymond Douglas [1]

## Abstract

This position paper argues that the dominant paradigm of AI evaluation (which focuses on superhuman autonomous performance and so implicitly targets the goal of replacing humans) is guiding AI development in the wrong direction. Instead, the AI community should pivot to evaluating the performance of human–AI teams. We argue that this collaborative shift will foster AI systems that act as true complements to human capabilities and therefore lead to far better societal outcomes than will the current process.

## 1. Introduction

By convention, in ML research we typically consider a problem 'solved' when the best system *autonomously* surpasses a human baseline (Wei et al., 2025). AI progress has largely been fuelled by this autonomous-competitive paradigm, of benchmarks that pit solo AIs against solo humans, with the implicit expectation that a successful system will work alone and replace human effort. While this **Replacement Paradigm** has driven innovation on many tasks, we argue that the AI research community must reorient AI evaluation, shifting from the current focus on benchmarks that measure solo AI against solo human performance—with the tacit aim of replacing humans at each task—to a new emphasis: evaluating the *collaborative* intelligence of human–AI teams. We argue that this is necessary for systems to gain crucial prosocial capabilities (Section 2.2).

The current emphasis on human-replacement metrics, while useful for cheaply gauging certain AI capabilities, inadvertently steers development towards AI systems as substitutes for human labor and cognition (Manheim & Garrabrant, 2018). This trajectory risks exacerbating economic inequalities, devaluing human skills, and ultimately leaving hu-

[1]ACS Research, CTS, Charles University, Prague, Czech Republic [2]Leverhulme Centre for the Future of Intelligence, University of Cambridge, Cambridge, UK. Correspondence to: Jan Kulveit <jk@acsresearch.org>.

*Proceedings of the 43rd International Conference on Machine Learning*, Seoul, South Korea. PMLR 306, 2026. Copyright 2026 by the author(s).

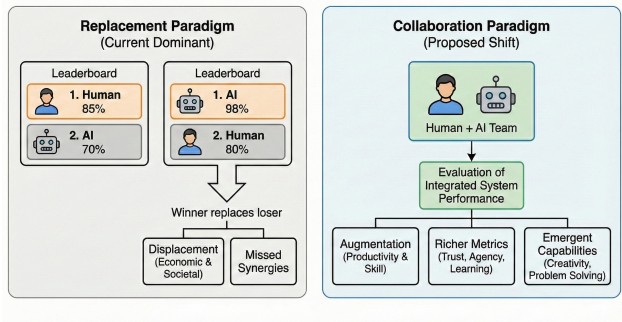

*Figure 1.* Schematic illustration of the proposed paradigm shift: from evaluating AI systems in isolation against human baselines (left) to evaluating human–AI teams working collaboratively toward shared goals (right).

manity disempowered (Autor et al., 2015; Brynjolfsson & McAfee, 2017; Jin et al., 2025; Kulveit et al., 2025). Furthermore, it often fails to capture or promote progress on the broader capabilities essential to real-world productivity, such as asking questions, building shared understanding, and the ability to foster human creativity or insight (Kamar, 2016; Braun et al., 2023; Schmutz et al., 2024).

This paper advocates for a shift towards evaluating human-AI collaborations, where the primary question is not "*Can an AI do this task instead of a human?*" but rather "*How much more effectively can a human (or team) achieve a goal in concert with an AI?*" We will explore the limitations of the current evaluation paradigm, articulate the benefits of focusing on human-AI team performance, propose key metrics and research directions for this new focus, address potential counter-arguments, and conclude with a call to action for the community to spearhead the reorientation. By prioritizing the development and evaluation of AI that enhances human agency and collective intelligence, we can guide AI's trajectory towards more beneficial and humane futures.

## 2. The Limits of the Replacement Paradigm

The prevailing paradigm in AI evaluation, characterized by its focus on beating human performance on discrete tasks, has undeniably enabled rapid advancements in algo-

rithmic capabilities (Russakovsky et al., 2015). Benchmarks across diverse domains—from natural-language understanding (e.g., GLUE (Wang et al., 2018), SuperGLUE (Wang et al., 2019), MMLU (Hendrycks et al., 2021)) to complex game-playing (e.g., Go (Silver et al., 2016), StarCraft (Vinyals et al., 2019))—often use human performance as the primary yardstick, celebrating systems that operate autonomously and, ideally, achieve superhuman results. While instrumental in demonstrating AI's potential, this human-replacement focus carries significant limitations and potential perils that warrant a critical reevaluation.

## 2.1. Economic and Societal Misalignments

A primary concern is the socioeconomic trajectory fostered by an evaluation framework that implicitly or explicitly prioritizes AI as a substitute for human labor. Economic theory suggests that technologies can act as either complements to or substitutes for labor (Acemoglu & Restrepo, 2018; 2019). Technologies that complement human skills tend to increase human productivity, create new tasks, and can lead to higher wages and employment. Conversely, technologies that primarily substitute for human labor can displace workers, depress wages for skills that become automatable, and exacerbate income inequality (Autor et al., 2015; Brynjolfsson & McAfee, 2017).

Benchmarks and evaluation steer AI progress (Russakovsky et al., 2015). By predominantly benchmarking AI in a manner that emphasizes its capacity to perform tasks *instead of* humans, the research community inadvertently steers innovation towards substitutive applications. This can lead to an "automation race" where the primary goal becomes matching human output at lower cost, rather than exploring how AI can empower humans to achieve more or tackle entirely new challenges (Jin et al., 2025). The societal consequences of widespread labor displacement, without commensurate creation of new, human-centric roles or robust social safety nets, are a significant concern (Kulveit et al., 2025) that an evaluation paradigm focused on human replacement fails to address. Indeed, focusing on how AI can augment human capabilities may lead to more widely shared prosperity and a more optimistic vision for the future of work.

## 2.2. A Narrow Conception of AI Progress

The human-replacement focus, while providing clear, quantifiable metrics, can lead to a narrowed conception of what constitutes "AI progress." Success is often equated with outperforming humans on existing tasks, potentially neglecting other dimensions of intelligence, particularly those crucial for effective interaction and collaboration. For instance:

- *Interpretability and explainability for users:* While explainable AI (XAI) is a well-established field, these benchmarks rarely evaluate the quality or utility of explanations from the perspective of a human collaborator trying to understand, trust, or debug an AI's behavior within a shared task.

- *AI inquisitiveness and expression of uncertainty:* Systems that can identify gaps in their own knowledge, ask clarifying questions, or effectively express uncertainty are crucial for robust human–AI teaming (Li et al., 2011; Zhang & Choi, 2025; Bansal et al., 2021); yet current benchmarks mostly reward confident, definitive answers.

- *Adaptability to human intent:* Effective collaborators must adapt to partners' cognitive state, expertise, and goals, but mainstream benchmarks seldom assess this adaptability.

- *Fostering human learning:* An AI partner could ideally help humans learn or gain deeper insights, yet few benchmarks incentivize the development of mentor-like systems.

Empirical studies corroborate these concerns: Bansal et al. show that maximizing standalone AI accuracy can paradoxically *decrease* overall human-AI team utility because humans struggle to predict or calibrate to highly complex models (Bansal et al., 2021). This narrow focus risks developing AI systems that are "brilliantly competent" in isolation but "socially inept" or difficult to integrate into complex human workflows where collaboration and shared understanding are paramount (Braun et al., 2023; Schmutz et al., 2024).

## 2.3. Overlooking Emergent Risks and Benefits of Human–AI Systems

A singular focus on autonomous AI performance can also lead to overlooking both emergent risks and unique benefits that arise specifically from human-AI interaction. As an example of good practice, evaluations of "dual-use" or "dangerous capabilities" (e.g., misuse for bioweapon design or misinformation (OpenAI, 2023)) are, in essence, evaluations of a human-AI team, albeit with a focus on negative outcomes. Focusing on the AI alone would greatly underestimate the risks.

Conversely, synergistic benefits—such as the creative breakthroughs observed when humans collaborate with AI in cooperative settings like Hanabi (Siu et al., 2021) or in large-scale collective intelligence contexts (Cui & Yasseri, 2024)—would be entirely missed if AI is only tested in isolation. An evaluation framework centered on human-AI teams is better positioned to identify and navigate these interaction effects.

# 3. What Human+AI Evaluation Could Look Like

To counteract the limitations of a purely human-replacement paradigm, we advocate for a significant shift towards the evaluation of human-AI teams, or "cyborg" systems. This approach does not seek to diminish the importance of understanding standalone AI capabilities but rather to complement them by assessing how effectively AI can work *with* humans to achieve shared goals. The central question becomes: How much more capable, efficient, insightful, or creative can a human partnered with an AI be, and what AI attributes foster such synergistic collaboration?

## 3.1. Defining Human+AI Team Evaluation

Human+AI team evaluation assesses the performance, interaction dynamics, and emergent capabilities of a combined system comprising one or more human users and one or more AI agents working towards a common objective. Unlike benchmarks that isolate the AI component, this paradigm explicitly considers the human as an integral part of the system being evaluated. The focus is on the *synergy* achieved: can the team accomplish what neither the human nor the AI could achieve alone, or achieve it significantly better? Recent work in cooperative game settings showcases that such evaluations are possible (Siu et al., 2021).

## 3.2. Key Metrics for Human+AI Teams

Evaluating human-AI teams requires a richer set of metrics beyond simple task accuracy or speed. We propose a three-pronged approach, summarized in Table 1.

## 3.3. Fostering Undervalued AI Capabilities

A shift towards human-AI team evaluation would naturally incentivize capabilities that current benchmarks undervalue (e.g. the skill of questioning, prompting a human collaborator in the most fruitful way; scaffolding human learning; stimulating curiosity and exploration; or mutual explainability). By focusing on these richer criteria, we can guide AI development towards systems that are not just powerful, but also effective, trustworthy, and empowering partners for humanity.

## 3.4. Not throwing the Waymo under the bus

Note that we do not argue for *abandoning* standalone evaluation—our contention is rather that it answers a different question than teaming evaluations, and that the field often implicitly treats it as the only question worth asking.

Autonomous driving is a good example: here, the system absolutely needs robust, standalone evaluation. But the deployment context is a human-AI system: human drivers

*Table 1.* Proposed metrics for evaluating human+AI team performance across three complementary dimensions.

| Metric | Description |
|---|---|
| *Dimension 1: Team Task Performance* | |
| Outcome Quality | Accuracy, creativity, and utility of results |
| Efficiency | Time and resources expended |
| Robustness | Performance stability across conditions |
| Innovation | Novelty of solutions produced |
| *Dimension 2: Human-Centric Outcomes* | |
| Satisfaction | User experience (e.g., SUS (Brooke, 1996), USE (Gao et al., 2018)) |
| Skill Enhancement | Learning and understanding gains |
| Cognitive Load | Mental effort management (Sweller, 1988) |
| Trust Calibration | Appropriate reliance on AI (Lee & See, 2004; Schmutz et al., 2024) |
| Agency | Sense of control and empowerment |
| *Dimension 3: Collaborative Fluency* | |
| Interaction Efficiency | Communication overhead and turn-taking |
| Shared Awareness | Common ground and mutual understanding (Braun et al., 2023) |
| Adaptability | Responsiveness to partner's needs |
| Error Resilience | Recovery from mistakes and misunderstandings |

share the road, operators monitor fleets, regulators set policy, and drivers are expected to take over when the system reaches its limits.

We are instead proposing three tiers of evaluation:

1. Standalone benchmarking, for basic capability screening — fast, cheap, scalable.

2. Surrogates of team evaluation for rapid iteration on collaboration design — moderate cost, reasonable scale.

3. Real-participant team evaluation, for deployment-critical assessment — slower and more expensive, but necessary for full validity.

Clearly not every model iteration needs tier 3 evaluation; that tier is reserved for systems approaching deployment (or for establishing a baseline team performance in a new domain).

# 4. Related Work

## 4.1. Human–AI Teaming in Machine Learning

Early "hybrid-intelligence" systems showed that dynamically routing subtasks between algorithms and crowd workers can outperform either party alone, e.g. (Kamar, 2016),

(Lasecki et al., 2011)'s *Legion* framework for real-time crowdsourced control, or (Amershi et al., 2014)'s interactive machine-teaching loop. Subsequent work generalised the pattern: in medical imaging, 'double-reading' pipelines pair a CNN with a radiologist to reduce false negatives while preserving throughput (McKinney et al., 2020); in games, agents explicitly optimised for cooperative play (e.g. Carroll et al. (2019) in Overcooked) achieve far higher human–AI team scores than self-play-only baselines. Parallel strands explore explanation and uncertainty communication as levers for collaboration—see (Ribeiro et al., 2016)'s LIME, (Senoner et al., 2024), or (Bhatt et al., 2021)'s calibrated confidence displays.

Recent empirical work has begun to reveal the nuanced reality of human-AI collaboration. Chen and Chan (Chen & Chan, 2024) examined different collaboration modalities with LLMs in creative work, comparing "ghostwriter" modes (where LLMs assume the main content generation role) versus "sounding board" modes (where LLMs provide feedback on human-created content). They found that using LLMs as sounding boards helped non-experts achieve content quality closer to experts, while using LLMs as ghostwriters was detrimental to expert users due to anchoring effects. Similarly, Kumar et al. (Kumar et al., 2024) investigated the long-term effects of LLM use on human creativity, emphasizing the importance of designing AI tools as "coaches" rather than "steroids" or "sneakers" to prevent stifling human creativity even after AI assistance is removed.

The challenge of evaluating human-AI collaboration has prompted new methodological frameworks. Fragiadakis et al. (Fragiadakis et al., 2024) developed a comprehensive framework proposing a structured decision tree to select relevant metrics based on distinct collaboration modes. Woelfle et al. (Woelfle et al., 2024) demonstrated that human-AI collaboration in evidence appraisal achieved accuracies of 89-96% for systematic review assessment, outperforming either humans or AI alone. Sharma et al. (Sharma et al., 2024) introduced a novel framework for measuring AI agency in collaborative tasks, based on social-cognitive theory. An influential manifesto for healthy collaboration with AIs is (Dupuis & Janus, 2023).

However, challenges remain in realizing the potential of human-AI teams. Schmutz et al. (2024) found that adding an AI teammate often reduces coordination, communication, and trust, with trust in AI tending to decline over time due to initial overestimation of its capabilities.

Despite this rich literature, the dominant framing remains performance-motivated: collaboration is valued mainly insofar as it lifts headline metrics. Benchmarks still rank agents in isolation, and so attributes like fostering user learning, sustaining calibrated trust, or preserving human agency are rarely optimised. Our position paper extends this body of work by arguing that *why* we pursue teaming—and how benchmarks steer research incentives—matters as much as raw task gains.

One major exception is Haupt & Brynjolfsson (2025), which provides an economic framework in support of a similar argument to ours. We differ in emphasis: they seek a sustainable economic setup where human labour remains relevant, while we additionally view cyborg evaluations as directly promoting prosocial AI capabilities. Some complementary ideas provided by Haupt & Brynjolfsson include the use of crowdworkers, running RCTs to causally identify the value of changing the inputs to the 'centaur production function', and persistent leaderboard competitions.

A concrete example of progress towards cooperative evaluation is the Docent tool (Meng et al., 2025), which helps human evaluators to understand the vast logs produced by AI agents, identify anomalous reasoning and behavior, and then intervene on the target system.

Recent editions of the Anthropic Economic Index (Appel et al., 2026) report on some collaborative metrics, including the proportion of tasks automatically classified as human-augmenting (currently 52%, up from 47% in 2025), and the maximum time horizon of successful tasks in real user sessions. This metric thus *tacitly* measures the performance of a human+AI system, since the user is selecting tasks suitable for the AI, providing correction to AI outputs, solving subtasks, and generally steering the task towards success over multiple turns.

## 4.2. Technology, Automation, and Labor Economics

Task-based models in economics formalise two opposing forces: 1) automation displaces workers by letting capital perform existing tasks, while 2) innovation that creates *new* human-advantageous tasks reinstates demand for labour. For instance, Autor (Autor, 2015) documents how the IT wave of automation hollowed out routine jobs, yet boosted demand for abstract and service work. Acemoglu & Restrepo quantify the displacement/reinstatement balance and warn that innovation which instead skews toward further substitution can suppress both wages and productivity growth (Acemoglu & Restrepo, 2018; 2019). A broader coalition of economists and computer scientists has argued that the dominant vision of autonomous AI "misconstrues the nature of intelligence," which is fundamentally social and relational, and that AI development should instead prioritize augmenting human creativity and cooperation (Siddarth et al., 2022).

Complementary technologies—what Brynjolfsson & Mitchell call 'augmentation'—tend to raise productivity *and* employment, but the direction of technical change is endogenous to incentives (Brynjolfsson & Mitchell, 2017).

Existing models treat that direction as an aggregate choice driven by factor prices or policy; they seldom probe the micro-level mechanisms—publication norms, leaderboards, benchmark design—that channel frontier ML research.

By proposing benchmarks that measure the *synergy* and value-to-humans of AI, we provide a concrete lever to shift those incentives. Our argument thus links the ML teaming literature's empirical insights with macroeconomic discourse on *augmentative* versus *automating* innovation.

## 5. Alternative Views

Our proposition to shift the target of AI evaluation towards human-AI team performance faces some natural challenges:

### 5.1. Cost and Complexity of Human Involvement

Clearly, one reason for the dominance of the autonomous-competitive style of evaluation is simple cost: it is far slower and more expensive to gain IRB approval, recruit, coordinate, and measure human performance relative to a static dataset benchmark. So bringing humans in could slow the rapid iteration cycles currently normal in AI research.

**Rebuttal:** While acknowledging the increased logistical demands, we argue for the following mitigating factors and overriding benefits:

- *Strategic investment:* The potential gains in developing AI that truly augments human capability and integrates safely into society justify the investment.

- *Amortizing costs with shared infrastructure:* The community can develop shared platforms, standardized interactive environments (e.g., "Human-AI Interaction Gyms"), and best practices to streamline human-AI evaluation. The human labour for such evaluations can be crowdsourced or supplied in a commoditized way. We note the precedent of the successful crowdsourced LMArena leaderboard (Chiang et al., 2024) and that the clinical trial field has developed a sophisticated market for fast outsourcing of participant recruitment (Mirowski & Van Horn, 2005).

- *Human surrogate models:* A promising research avenue is the development of AI-based 'human surrogate models' (Kofler et al., 2022; Anthis et al., 2025). These models, trained on data from human interactions, could enable more rapid and scalable iteration for many aspects of collaborative AI development.

- *Phased and tiered evaluation:* Not all human-AI evaluations need to be large-scale. Simpler, more constrained interactive tasks can be used for initial assessments, with more complex evaluations reserved for systems showing promise.

A challenge for current LLM-based human surrogates is that they likely underrepresent cognitive biases, fatigue effects, emotional states, and the full diversity of real users (Hullman et al., 2026). We thus propose that surrogates be used mainly for rapid screening and preliminary iteration, but not as a substitute for evaluation with real participants. However, we expect that the more researchers use proxies, the more incentive there will be to study the efficacy of these proxies and so direct attention to improving the simulations.

### 5.2. Not 'Pure' AI Research

It could be argued that our proposed focus on human-AI interaction shifts AI research away from 'core' challenges and towards Human-Computer Interaction (HCI) and human-factors engineering.

**Rebuttal:** Building AI capable of effective human collaboration is not a pure AI problem, but is instead a multidisciplinary problem with a large ML research component.

- *New AI frontiers:* Achieving genuine human-AI synergy would benefit from progress in areas such as adaptive AI, explainable AI tailored for collaborators, representational alignment, AI alignment and novel human feedback paradigms. These are deeply technical AI research problems.

- *Intelligence in context:* True intelligence, whether artificial or natural, is often best understood and expressed through interaction.

### 5.3. This Is Just HCI

It could be said that the above argument has a simple solution: ML people should just use existing methods from the field of human-computer interaction.

The biggest challenge in transferring methods developed for (users of) ordinary software to the AI context is the validation of proxy measures at scale. While HCI has validated instruments for trust (Schmutz et al., 2024), cognitive load (Sweller, 1988), shared awareness (Braun et al., 2023), etc., they require surveys, think-aloud protocols, or trained observers (Zhang & Zhang, 2019) –and none of these validations scale to leaderboard conditions (with e.g. hundreds of remote and often anonymous submissions).

The part of our problem most foreign to HCI methods is integrating humans into the training loop of AIs. Achieving this also depends on us solving proxy validation: one needs to know that one's log-derived signal actually measures what one claims, before one optimises against it.

However, we think that solving this proxy problem is not a blocker to our programme — if we have strong benchmarks for measuring human+AI team performance, then

this will itself create an incentive to find ways to improve team performance. These could include new collaboration-promoting training methods, but, equally, they could instead look like model scaffolding, fast steering of inference, or dataset curation.

### 5.4. Human Variability and Reproducibility

Human participants introduce variability (in skill, motivation, strategy, etc.) that can make benchmarks noisy and results hard to reproduce. Clearly this counts against one core tenet of scientific progress (Donoho, 2024).

**Rebuttal:** While human variability is undeniable, established research methodologies can address this, and the challenge itself can spur innovation.

- *Robust experimental design:* Methodologies from human-subjects research (e.g., within-subject designs, appropriate sample sizes, clear reporting of participant characteristics, standardized protocols) can mitigate and account for variability.

- *AI adaptability:* The evaluation can focus on the AI's ability to adapt to different users or to improve team performance over time with a specific user, turning variability into a feature to test against.

- *Human Surrogate Models (Revisited):* Validated human surrogate models can offer a less noisy baseline for comparing certain AI capabilities, complementing direct human studies.

- *Multifaceted metrics:* Relying on diverse metrics can provide a more complete and robust picture than a single potentially noisy score.

A particularly important confounding variable in human+AI team evaluation is the skill level of the sampled humans at working with AI (Wang et al., 2023). Being able to measure this is thus on the critical path for our proposed collaborative evaluation, and so is a good place to start.

### 5.5. The Subjectivity of Good Collaboration

What constitutes "good" or "effective" collaboration can be subjective and harder to quantify than objective task performance like accuracy.

**Rebuttal:** While perfect objectivity is elusive, meaningful and actionable metrics for collaboration quality are achievable:

*Operationalization:* We can operationalize aspects of good collaboration (e.g., efficient communication, error detection rate, shared attention) into observable behaviors.

*Validated subjective measures:* Standardized and validated questionnaires from HCI and psychology can reliably measure subjective experiences like satisfaction, cognitive load, and trust.

*Combining objective and subjective data:* A combination of objective task outcomes and subjective process measures provides a comprehensive view of collaborative efficacy.

### 5.6. Collaboration is not monolithic

Attempting a single framework that covers all potential human+AI collaborations is hubristic:

**Rebuttal**: We agree in spirit, but simply note that the metrics listed in Table 1 can easily be selected and weighted differently based on the specific collaboration being tested. We build on Fragiadakis et al. (2024) by proposing the following types of collaboration, split by the temporal nature of the interaction, and corresponding metrics: 1) real-time co-creation (e.g., pair programming): here the metrics prioritised would be fluency, turn-taking skill, and the maintenance of 'flow'; 2) asynchronous assistance (e.g., AI drafting a doc, human then editing it): prioritise output quality delta, ease of revision; 3) decision support (e.g., AI recommends, human decides): would prioritise calibration, override quality, agency; 4) oversight and monitoring (e.g., AI decides, human supervises): prioritise error detection rate, attention sustainability.

## 6. Call to Action

We call upon the reader to:

- *Publish human–AI interaction datasets:* Release annotated logs of collaborative sessions, including both successful and failed interactions, to enable research on team dynamics.

- *Develop standardized evaluation infrastructure:* Create shared platforms and APIs ("Human–AI Interaction Gyms") that reduce the overhead of running collaborative benchmarks.

- *Report collaborative metrics alongside solo benchmarks:* When releasing new models, include evaluations of how effectively humans can work with them, not just autonomous performance.

- *Invest in human surrogate models:* Develop validated simulations of human collaborators to enable rapid iteration before costly human studies.

A first evaluation of human+AI teaming could look roughly as follows: beginning in the software engineering domain, build interactive versions of existing standalone benchmarks

(e.g. Terminal-Bench) and enrich the existing form of 'up-lift' studies (Paskov et al., 2026). A minimal viable teaming benchmark could be a set of real GitHub issues drawn from open-source repositories, with study participants working in one of four conditions (human alone; AI alone; human+AI synchronous; human+AI asynchronous). The initial metrics could be drawn from our Table 1: task performance delta (in issues resolved, code quality via test coverage and review acceptance), efficiency (time to resolution), trust calibration (frequency and correctness of accepting vs. over-riding AI suggestions), and skill enhancement (unassisted performance on a held-out problem set before and after the AI-assisted session). This domain is well-suited for a first step because the task outcomes are relatively objective, interaction logs are naturally generated, and there is a large existing population of developers with measurable skill variation.

## 7. Conclusion

We believe that the challenges listed above, while real, are surmountable. Moreover they are in fact exciting research opportunities, and are of a similar scale to those the field has risen to in the past. Anyway the imperative to develop AI that works *with* humans outweighs the difficulties associated with evolving our evaluation paradigm.

## Acknowledgments

This work was supported by the Czech Science Foundation, grant No. 26-23955S.

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
