# OpenReview forum: "Position: AI Evaluation Should Work With Humans"
_ICML.cc/2026/Position_Paper_Track — ICML 2026 Position Paper Track regular_

### Official Review · Reviewer_WJgz · 2026-03-05

**Significance:** 3
**Argument Clarity:** 2
**Rating:** 3
**Confidence:** 4

**Questions:**

(1) Is it possible to propose one minimal viable human–AI team benchmark (one task domain + standard protocol + metrics + baselines)? Or any existing work can provide insight?

(2) Given the substantial HCI/human-factors literature on human–AI collaboration evaluation, where exactly is the biggest gap when transferring these methods into ML’s benchmark/leaderboard ecosystem (e.g., scalability, task representativeness, tight training–evaluation loops, incentive design)?

**Alternative Views Section:**

Yes

**Compliance With Llm Reviewing Policy A Conservative:**

Affirmed.

**Discussion Potential:**

3

**Final Justification:**

Thank authors for the rebuttal. On balance, I appreciate the rebuttal and have higher confidence in the authors’ framing, but I do not think it fully changes my overall assessment of the current submission. I maintain my score.

**Paper Summary:**

This position paper argues that evaluation in AI should shift from replacement-style metrics (how well a model performs instead of a human) to human–AI team performance (how well humans and AI collaborate). The authors propose a high-level framework that groups team-evaluation measures into three categories: task outcomes, human-centered outcomes, and collaboration fluency. They motivate the shift by noting that evaluation paradigms shape research incentives and deployment choices, and they discuss common objections to human-in-the-loop evaluation—cost, reproducibility, and subjectivity—along with mitigation directions such as shared evaluation infrastructure, tiered evaluation pipelines, and the use of human surrogate models to improve scalability.

**Position:**

Yes

**Position In Title:**

Yes

**Related Work:**

2

**Strengths And Weaknesses:**

Strengths:

The paper moves from limitations of current evaluation practice to a proposed alternative framing (definitions + metric categories), then anticipates counterarguments and outlines action-oriented paths forward, making it easy to follow and likely to spur discussion.

Weaknesses:

(1) Lacks a concrete “first-step” pathway: The proposal reads more like a directional call with metric lists than an actionable roadmap (which task domains to prioritize, how to standardize protocols and control cost/reproducibility, and what a minimal viable benchmark/leaderboard would look like in practice).

(2) Related HCI/human-factors evaluation work is cited, but the paper could more systematically explain what existing HCI toolkits already provide for human–AI collaboration evaluation and what new contribution is needed for ML’s benchmark culture (e.g., scalability, task representativeness, linkage to training loops, leaderboard incentives).

(3) Core constructs remain broad, weakening testability: Dimensions like agency, shared awareness, or innovation are important but can be operationalized very differently across tasks; clearer must-have vs optional dimensions and recommended measurement/reporting templates would help.

(4) Boundary with standalone evaluation could be sharper: The paper acknowledges the value of offline/standalone eval, but could more precisely articulate division-of-labor and integration (what can remain offline vs what must be evaluated in interactive team settings) and how to avoid team-eval becoming an unscalable user-study bottleneck.

**Support:**

3

---

> ### Author Rebuttal · Authors · 2026-03-31
>
> We appreciate this reviewer’s detailed engagement and their clear articulation of what the paper needs to become more actionable. We address each weakness and question below.
>
> **W1/Q1: Lacks a concrete “first-step” pathway.**
>
> We agree this was a gap. Here is a concrete sketch. We propose software engineering as the first domain, building on existing standalone benchmarks (SWE-bench) and uplift studies (METR 2025). A minimal viable team benchmark: a set of real GitHub issues drawn from open-source repositories, with participants working in four conditions (human alone, AI alone, human+AI synchronous, human+AI asynchronous). The initial metrics, drawn from our Table 1: task performance delta (issues resolved, code quality via test passage and review acceptance), efficiency (time to resolution), trust calibration (frequency and correctness of accepting vs. overriding AI suggestions), and skill enhancement (unassisted performance on a held-out problem set before and after the AI-assisted session). This domain is well-suited because task outcomes are relatively objective, interaction logs are naturally generated, and there is a large existing population of developers with measurable skill variation. We fold this sketch into the revision.
>
>
> **W2/Q2: Should explain what existing HCI toolkits provide vs. what ML’s benchmark culture needs.**
>
> We agree this gap is underexplored. In our view, the biggest challenge in transferring these methods is probably proxy validation at scale. HCI has validated instruments for trust, cognitive load, shared awareness etc — but they require surveys, think-aloud protocols, or trained observers, none of which scale to leaderboard conditions. The core research program is systematically validating log-derived behavioral proxies against these gold-standard measures. For instance: does the pattern of accepting, modifying, or rejecting AI suggestions correlate reliably with trust calibration as measured by established scales?
>
> Reflecting on some of the other examples offered, we think incentive design is best viewed as an adoption barrier rather than a methodological one: it applies to any evaluation reform and is downstream of having methods worth adopting. Similarly, task representativeness is real but not specific to team evaluation; standalone benchmarks face the same ecological validity questions, and progress there (e.g., the shift from synthetic tasks to real-world samples like SWE-bench) transfers directly.
>
> Training loop integration does seem like a real gap and the one most foreign to HCI. But it depends on solving proxy validation first: you need to know your log-derived signal actually measures what you claim before you optimise against it. However, we think that solving this is less of a priority — if there are clear benchmarks for team performance, then this will create an incentive to find ways of improving team performance. This could look like training methods, but equally it could look like scaffolding or inference approaches, or dataset curation.
>
> **W3: Core constructs are too broad. Need must-have vs. optional dimensions.**
>
> We propose three must-haves: task performance delta, efficiency, and calibrated override rate. These are measurable today with existing methods, are domain-agnostic, and directly capture the human+AI delta that is invisible under standalone evaluation. The remaining metrics are important but context-dependent: some domains will need innovation measurement, others will prioritise cognitive load or skill enhancement.
>
> **W4: Boundary with standalone evaluation could be sharper. Risk of team-eval as bottleneck.**
>
> We envision a tiered pipeline:
> - Standalone benchmarks for basic capability screening — fast, cheap, scalable.
> - Surrogate-based team evaluation for rapid iteration on collaboration design — moderate cost, reasonable scale.
> - Real-participant team evaluation for deployment-critical assessment — slower and more expensive, but necessary for validity.
>
> Not every model update needs tier 3; that tier is reserved for systems approaching deployment or for establishing baseline team performance in a new domain. Standalone evaluation remains the workhorse for capability development; surrogate testing is fairly cheap on the margin; team evaluation is the gate for deployment readiness.

---

> > ### Author Rebuttal · Reviewer_WJgz · 2026-04-03
> >
> > Thank authors for the rebuttal. On balance, I appreciate the rebuttal and have higher confidence in the authors’ framing, but I do not think it fully changes my overall assessment of the current submission. I maintain my score.

---

### Official Review · Reviewer_vJjc · 2026-03-09

**Significance:** 3
**Argument Clarity:** 3
**Rating:** 4
**Confidence:** 4

**Questions:**

1.Within your proposed three-layer framework of metrics, which ones currently have relatively mature measurement methods that could be applied simply by shifting the evaluation subject? Which ones urgently require methodological innovation?

2.How can we ensure that AI-simulated "human proxies" do not overlook the irrational decision-making and emotional fluctuations of real humans in collaboration?

3.In Table 1, if there is a conflict between "Efficiency" and "Skill Enhancement" (i.e., a human becomes slower in order to learn), how should the evaluation framework balance the weighting?

**Alternative Views Section:**

Yes

**Compliance With Llm Reviewing Policy A Conservative:**

Affirmed.

**Discussion Potential:**

2

**Final Justification:**

The paper raises an important and timely point about shifting from standalone evaluation to human–AI team evaluation, and I find the motivation and framing compelling, especially in highlighting dimensions like agency, collaboration, and long-term system performance that are often overlooked. My initial concerns were around whether the argument sufficiently justified not prioritizing the traditional paradigm, the feasibility of measuring proposed metrics, and the lack of differentiation across collaboration settings. The rebuttal addresses these points reasonably well: it clarifies that standalone and team evaluation are complementary rather than competing, provides a more nuanced breakdown of metrics (including which are mature versus exploratory), and introduces a useful taxonomy of collaboration modes that strengthens the conceptual clarity. The discussion of human proxies and trade-offs is also more balanced. That said, the work remains somewhat high-level, and many of the proposed evaluation dimensions still lack concrete, scalable methodologies. Overall, the rebuttal improves my confidence in the paper and addresses several key concerns, but some limitations in concreteness and validation remain, leading me to maintain a borderline accept recommendation.

**Paper Summary:**

This paper challenges the convention of treating "surpassing human baselines" as task completion. It argues that replacement-oriented evaluation leads to socio-economic risks and neglects AI progress in interpretability, uncertainty expression, and human augmentation. The authors propose a "Collaboration Paradigm" evaluating human-AI teams as integrated systems , introducing a multi-dimensional metric matrix and a call to action for shared infrastructure.

**Position:**

Yes

**Position In Title:**

Yes

**Related Work:**

2

**Strengths And Weaknesses:**

Strengths:

1.The topic (evaluation paradigms) is a core meta-question for the ML community. The argument for rethinking evaluation goals is highly relevant as AI capabilities approach or surpass human baselines in various domains.

2.Effectively grounded in labor economics and HCI theories.

3.Provides a concrete, multi-dimensional metric matrix (Table 1).

Weaknesses:

1.The paper provides less discussion on "why the replacement paradigm should not be abandoned entirely." In certain safety-critical domains (e.g., autonomous driving), ensuring the AI's independent and reliable performance in edge cases may take precedence over the collaborative experience. Acknowledging the continued necessity of the "replacement paradigm" in certain sub-fields could make the paper's position appear more balanced and robust.

2.The evaluation metrics proposed in the paper (e.g., "innovativeness," "human agency," "shared awareness") present significant measurement challenges. The paper could benefit from a deeper discussion on the existing measurement methods for 1-2 core metrics and how they could be adapted within the new paradigm.

3.Insufficient consideration of the definition and contextual diversity of "collaboration": The paper proposes "human-AI collaboration" as a monolithic concept but does not delve into how collaboration and its evaluation priorities should differ vastly across task types (e.g., creative generation vs. high-stakes decision-making) and interaction modes (e.g., real-time co-creation vs. asynchronous assistance).

**Support:**

3

---

> ### Author Rebuttal · Authors · 2026-03-31
>
> We thank this reviewer for their thoughtful and specific feedback, particularly the attention to measurement challenges and the diversity of collaboration contexts.
>
> **W1: Insufficient discussion of why the replacement paradigm should not be abandoned entirely (e.g., autonomous driving).**
>
> We don’t argue for abandoning standalone evaluation — our contention is that it answers a different question from team evaluation, and that the field often implicitly treats it as the only question worth asking. Autonomous driving is a good example: the vehicle's autonomous capabilities absolutely need robust standalone evaluation. But the deployment context is a human-AI system — human drivers share the road, operators monitor fleets, regulators set policy, and drivers are expected to take over when the system reaches its limits. Standalone evaluation tells you whether the AI component is capable; team evaluation tells you whether deploying it in a human context makes the overall system better. However, it is true that eventually this might cease to be the case. In the revision, we use this example to illustrate the complementary relationship.
>
> **W2/Q1: Metrics like innovativeness, human agency, shared awareness have significant measurement challenges.**
>
> Agreed. We distinguish between metrics with mature methods that straightforwardly extend to human+AI settings (outcome quality, efficiency, robustness, satisfaction, skill enhancement, cognitive load, trust calibration, error resilience) and those requiring methodological innovation in the context of human+AI collaboration (innovation, agency, interaction efficiency, shared awareness, adaptability). The revised Table 1 annotates this distinction. We believe it is important not to shy away from properties that are currently hard to measure — that friction is part of what has pushed the community toward paradigms that neglect them.
>
> And there is certainly existing HCI and psychology literature that can serve as a foundation for developing that methodology — taking the case of innovativeness, one could look at expert panel ratings or Torrance-style divergent thinking measures. Even if these are hard to scale, they can be used as a standard against which to test more scalable approaches.
>
> **W3: Collaboration treated as monolithic. Should differ across task types and interaction modes.**
>
> A fair criticism. We propose a taxonomy of collaboration modes with differentiated evaluation priorities. This is complementary to Fragiadakis et al. (2024), which organises modes by locus of control; ours organises by the temporal and structural properties of the interaction:
> * Real-time co-creation (e.g., pair programming): prioritise fluency, turn-taking, flow.
> * Asynchronous assistance (e.g., AI drafts, human edits): prioritise output quality delta, revision effort.
> * Decision support (e.g., AI recommends, human decides): prioritise calibration, override quality, agency.
> * Oversight and monitoring (e.g., human supervises AI): prioritise error detection rate, attention sustainability.
>
> This justifies why metric weightings should be context-dependent rather than universal.
>
> **Q2: How to ensure human proxies don’t overlook irrational decision-making and emotional fluctuations?**
>
> We share this concern. Current LLM-based human surrogates are likely to underrepresent cognitive biases, fatigue effects, emotional states, and the full diversity of real users. We propose that surrogates be used mainly for rapid screening and preliminary iteration, but not a substitute for evaluation with real participants. However, we expect that the more researchers use proxies, the more incentive there will be to study the efficacy of these proxies, so to some extent we think this problem will begin to solve itself if there is the will.
>
> **Q3: If Efficiency and Skill Enhancement conflict, how to balance weighting?**
>
> This is exactly the kind of trade-off our framework is designed to surface rather than resolve with a fixed weighting. The appropriate balance depends on context: a training environment for junior professionals should prioritise skill enhancement even at some efficiency cost; a crisis-response system should prioritise speed. Time horizon also matters here: short-run efficiency loss from learning may yield long-run efficiency gains as human capability improves. Under the replacement paradigm, this trade-off is hard to notice — the learning dimension is simply not measured, so the trade-off cannot even be articulated. Our framework makes it visible and subject to deliberate, context-sensitive judgement.

---

> > ### Author Rebuttal · Reviewer_vJjc · 2026-04-03
> >
> > The authors have addressed my concerns, but considering the overall quality of the paper, I recommend a borderline accept.

---

### Official Review · Reviewer_Pn6N · 2026-03-13

**Significance:** 4
**Argument Clarity:** 3
**Rating:** 5
**Confidence:** 4

**Questions:**

(1) “This approach does not seek to diminish the importance of understanding standalone AI capabilities”, if standalone systems outperform cyborgs, what should be the next course of action in this new paradigm of AI eval with humans?

(2) What is the position in this paradigm of determining which of these metrics are more important than the other? As mentioned in weakness (2), is there a principled way to aggregate across all three dimensions, or does the framework require human judgment to determine which technique wins?

**Alternative Views Section:**

Yes

**Compliance With Llm Reviewing Policy A Conservative:**

Affirmed.

**Discussion Potential:**

4

**Final Justification:**

I maintain my recommendation of accept.

**Paper Summary:**

The paper argues that rather than treating human capabilities as the baseline to beat, AI should be benchmarked against the combination of human+AI, moving away from replacing humans to augmenting their capabilities.

**Position:**

Yes

**Position In Title:**

Yes

**Related Work:**

3

**Strengths And Weaknesses:**

Strengths:
(1) The paper is well organized and the motivation and goals of the paper are clear.

(2) The position clarifies that it is distinct from prior works (Haupt & Brynjolfsson (2025))

(3) Their proposed type of metrics with the 3 complementary dimensions supports their arguments by extending the current way the ML community largely assesses AI systems.

(4) There is a comprehensive discussion of relevant literature, pertaining not only to human-ai performance but also to prior work in economics and automation that can inform how newer technology like AI can disrupt the workforce.
The paper does a great job of presenting the alternative views, discussing in depth the potential downsides for including humans (variability, costs, subjectivity), and presents thoughtful rebuttals.


Weakness:
(1) The paper lists proposed metrics to measure cyborg performance, most of these support the paper’s position, however some such as *Innovation* seem vague, how does one measure novelty? How would we create a benchmark for something like “Skill Enhancement”.

(2) Additionally, the paper proposes we benchmark with this set of metrics, but does not discuss the likely common scenario of a new model improving in *Team Task Performance*, and *Collaborative Fluency* but dropping performance in *Human-Centric Outcomes*.

**Support:**

4

---

> ### Author Rebuttal · Authors · 2026-03-31
>
> We are grateful for this reviewer’s generous and careful reading.
>
> **W1: Innovation is vague. How to measure novelty or benchmark Skill Enhancement?**
>
> We agree these need sharper treatment. Skill Enhancement is quite operationalisable: a pre/post design comparing unassisted human performance before and after AI-assisted work, relative to a control group.
>
> Innovation is harder — existing creativity assessments (expert panel ratings, Torrance-style divergent thinking measures) could be adapted but have not been validated in human–AI team settings. This is, we believe, an open problem. We keep it in the framework deliberately, because omitting hard-to-measure dimensions is one mechanism by which the replacement paradigm narrows what counts as progress. In the revised Table 1 we annotate each metric to distinguish what is measurable today from what requires methodological investment.
>
> **W2: Does not discuss trade-offs when a model improves Task Performance and Collaborative Fluency but drops Human-Centric Outcomes.**
>
> This is a fair point. We definitely do not want to give the impression that human+AI performance could be reduced to a single metric, and that is part of why we list and group a variety of factors. We add a brief discussion of this trade-off scenario in the revision, using it to illustrate why multi-dimensional reporting matters.
>
> **Q1: If standalone AI outperforms cyborgs, what is the next course of action?**
>
> This is a difficult question that breaks down into a few pieces. The natural answer may often be “replace humans with AIs” but that won’t always be the case.
>
> First, “outperforms” on what? If the metric is task accuracy alone, AI may well surpass human-AI teams in narrow, fully specified domains. Chess is the canonical case: centaur teams outperformed pure engines from roughly 2005 to 2013, but modern engines are strong enough that human involvement adds negligible value on the specific metric of game-winning. However, chess is the easy case for the replacement paradigm — it has complete information, an objective outcome, and a fully specified rule set. Most real-world domains are not like this.
>
> Second, even where AI alone outperforms on the primary task metric, team evaluation reveals dimensions that matter for deployment: Does the human retain the ability to function if the system fails? Are the AI’s error modes correlated with or complementary to human error modes? Does the AI’s advantage hold across edge cases and demographic subgroups? These questions are invisible under standalone evaluation.
>
> Third, the finding itself is informative: “AI alone beats the team” tells us the collaboration interface is adding friction rather than value, or potentially that humans are failing to trust AI outputs enough. So even the “negative” result from team evaluation is more useful than no team evaluation at all.
>
> Finally, there are domains where human involvement is required for accountability, legality, or trust regardless of relative performance. In these domains, evaluating the quality of that involvement is important regardless of whether the AI could in principle operate alone. It then becomes a difficult question about tradeoffs between performance and these other values.
>
> **Q2: Is there a principled way to aggregate across the three dimensions? Are there principled reasons to view some as more important or does it require human judgment?**
>
> We deliberately avoid proposing a single aggregate. We believe the three dimensions should be reported as a profile, with aggregation left to the decision-maker. One analogy here is medical intervention evaluation: efficacy, side effects, quality-of-life impact, and cost are reported separately — no one argues oncology needs a single number combining tumour response rate with nausea severity. Some deployment contexts will eventually need principled aggregation methods, but we think too much reduction right now would be premature.

---

> > ### Author Rebuttal · Reviewer_Pn6N · 2026-04-02
> >
> > I maintain my recommendation of accept.

---

### Official Review · Reviewer_XTVq · 2026-03-13

**Significance:** 3
**Argument Clarity:** 3
**Rating:** 5
**Confidence:** 2

**Questions:**

See Weaknesses section above.

**Alternative Views Section:**

Yes

**Compliance With Llm Reviewing Policy A Conservative:**

Affirmed.

**Discussion Potential:**

3

**Final Justification:**

The authors will have addressed my weaknesses and questions with their promised revisions and inclusion of the clarifications and responses to my questions in the next draft.

**Paper Summary:**

- The paper argues why AI evaluations should involve human-AI teams, towards building AI that complements (rather than replaces) humans.
- The paper discusses the limitations of current AI evaluation paradigms, benefits of evaluating human-AI teams, possible metrics, and challenges.

**Position:**

Yes

**Position In Title:**

Yes

**Related Work:**

3

**Strengths And Weaknesses:**

*Strengths:*
- The paper provides concrete evidence for the need to evaluate AI with human-AI teams in mind (e.g., devaluing human labor, narrow view of AI progress, underestimating AI risks).
- The paper connects AI evaluation paradigms to the concepts of complementary vs. substitutive technologies in economics.
- The paper provides a detailed discussion of related work on human-AI teaming and technological automation. The paper explicitly distinguishes itself as extending work on human-AI teaming by arguing (normatively) *why* evaluating human-AI teams is important, beyond human-AI complementarity.
- The paper addresses challenges with involving humans in evaluations, including scale and variability.
- The paper will spark important discussions about how to validly capture human-AI interactions in evaluations.

*Weaknesses:*
- The abstract does not offer sufficient detail about what topics the paper will cover (e.g., economic and societal displacement, metrics for evaluating human-AI teams).
- The paper could go into more detail about how the politics of current AI benchmarks (i.e., how benchmarks influence what tasks are considered valuable, and vice versa) reinforces the replacement paradigm.
- The paper should discuss more precisely how human+AI evaluations could complement AI-only evaluations (e.g., how might we measure uplift and what would positive vs. negative uplift entail for different types of tasks?).
- While hinted at in Section 5.4, the proposed metrics in Table 1 are task-agnostic but perhaps should depend on the type of task. For example, when a human uses AI to generate misinformation, the outcome quality and efficiency should be poor. If a human seeks validation in a harmful situation, an AI system should not prioritize human satisfaction.
- The paper should discuss the limitations of human surrogate models for simulating human interaction with AI, as well as considerations to ensure ethical and diverse participant recruitment.
- The calls to action could be more precise. For example: What might a possible schema for a human-AI interaction dataset look like? To what extent can we standardize human-AI interaction environments given human variability?

**Support:**

3

---

> ### Author Rebuttal · Authors · 2026-03-31
>
> We thank the reviewer for their specific pointers on how the paper can be improved, and we are glad they feel the paper can spark important discussions. Taking the weaknesses in turn:
>
> **W1: Abstract lacks detail on the paper’s contents.**
>
> Agreed — the revised abstract will preview our core contributions: metrics for complementarity, the grounding in replacement vs. substitution, and our calls to action.
>
> **W2: Lack of detail on how the politics of benchmarks reinforces the replacement paradigm.**
>
> We appreciate this suggestion. In our view, the mechanism is largely structural: benchmarks serve as coordination devices around measurable tasks, and the incentive ecosystem — funding tied to leaderboard ranks, publication norms that reward state-of-the-art claims, corporate investment in automation — locks in the replacement paradigm. Individual researchers face a collective-action problem: unilaterally adopting team evaluation is costly and unrewarded. This is precisely why community-level action (shared infrastructure, reporting norms) is necessary. We will incorporate this analysis in the revision.
>
> **W3: How might human+AI evaluations complement AI-only evaluations? How to measure uplift?**
>
> There are existing studies that attempt to measure uplift (e.g., the METR early-2025 study of AI-assisted software development), though it is methodologically quite tricky. Uplift depends heavily on the task, the user's expertise, and the interaction mode. In general, this makes it easier to underestimate the potential for collaboration, but we suspect this is partly because the evaluations tend to focus only on overall performance, measured the same way you measure standalone performance — which is indeed why we advocate for a variety of metrics.
>
> We discuss the implications of negative uplift specifically across different tasks in our response to Q1 from reviewer Pn6N, and we agree this is an important point that we incorporate more carefully into the revision.
>
> **W4: Metrics in Table 1 should be task-dependent. E.g., misinformation or harmful uses.**
>
> We agree that there is an important extra dimension in ensuring AI doesn't help people perform dangerous tasks — and incidentally, there are also some uplift studies here (e.g., OpenAI's work on biological threat creation) which have similarly faced methodological criticism. Our framework needs a normative layer for dual-use contexts, but developing that layer fully is beyond our current scope. The core goal of our paper is to show how evaluation norms push for human replacement, rather than to analyse where humans should (not) be uplifted. In the revised Table 1 we add brief caveats to make this scope clearer.
>
> **W5: Should discuss limitations of human surrogates and ethical/diverse recruitment.**
>
> Agreed. Looking over recent work, Gao et al. (2025, PNAS) find that LLMs fail to replicate human behaviour distributions even in simple strategic games. Work on LLM-simulated users in agentic evaluations finds they underrepresent demographic and cultural diversity (e.g. Seshadri 2026). However, a growing "simulate-then-validate" literature suggests surrogates can be useful for exploratory screening and hypothesis prioritisation (Hullman et al., 2026). We position surrogates as useful for rapid screening only, not as a substitute for real participants, and agree that participant diversity is especially important for team evaluation because collaboration quality can vary across user populations in ways standalone AI performance does not.
>
> **W6: Calls to action should be more precise. Schema for interaction datasets? Standardisation?**
>
> Premature standardisation carries real risks — baking in assumptions about which collaboration modes and metrics matter before the field has sufficient empirical grounding. Our revised paper distinguishes what can be productively standardised now (shared task environments, logging formats, a minimal set of certain reported metrics) from where the community should resist premature convergence (metric weighting, aggregation, normative judgments about acceptable trade-offs). This is analogous to how CONSORT standardises what clinical trials must report without prescribing what counts as a successful outcome.

---

> > ### Author Rebuttal · Reviewer_XTVq · 2026-04-03
> >
> > Thanks for your detailed and thoughtful response! I am happy to increase my score with the promised revisions and inclusion of the clarifications and responses to my questions in the next draft.

---

### Decision · Program_Chairs · 2026-04-30

**Decision:**

Accept (regular)

**Comment:**

** Overall: ** This position paper serves to argue that because benchmarks are designed to test AI models in a setting where they replace humans, we will only ever optimize for human replacement, even when this is undesirable. This topic has broad implications and significance to the ICML community, and reviewers agree that the paper is sufficiently detailed and opinionated to spur useful discussion.

** Primary strengths: **
- Broad significance: The argumentation draws on topics from a broad range of fields, including economics and HCI (XTVq, vJjc) and clearly distinguishes itself from prior work (Pn6N)
- Good discussion potential: How to capture human-AI interactions in eval is a widely applicable topic that will likely spark discussion at ICML (XTVq, WJgz), and the inclusion of concrete metrics and discussion will be beneficial to discussion (vJjc, XTVq)

** Primary weaknesses: **
- Unlear argumentation: The complementarity of AI-only vs. human-AI eval is not clearly discussed, and the limitations inherent to using humans, especially regarding human diversity/variance is not discussed in enough detail (XTVq). The authors reply in the rebuttal to agree that this is important, but this seems like a central point in incorporating humans in evals more broadly, and should be discussed with much clearer points.
- Unclear measurement possibilities: The paper does not sufficiently explain what metrics would/should apply to the proposal, and the use of the task-agnostic metrics in the paper will not always be appropriate (XTVq, vJjc). Some metrics are left especially vague, such as “innovation”, as it’s not clear how to measure something like novelty (Pn6N, WJgz). Additionally, the paper primarily considers performance changes for a single human interacting with an AI, but humans often work in teams, and so it follows that improving team performance is a reasonable and realistic target as well (Pn6N).
-  No clear roadmap or first steps towards implementing the proposal (WJgz). The authors agree to provide a sketch of an initial step in a revision, but the general point that the call to action would be more effective as a roadmap rather than a list of possible metrics remains.